# Objective Measurement of Subjective Pain Perception with Autonomic Body Reactions in Healthy Subjects and Chronic Back Pain Patients: An Experimental Heat Pain Study

**DOI:** 10.3390/s23198231

**Published:** 2023-10-03

**Authors:** Luisa Luebke, Philip Gouverneur, Tibor M. Szikszay, Wacław M. Adamczyk, Kerstin Luedtke, Marcin Grzegorzek

**Affiliations:** 1Institute of Health Sciences, Department of Physiotherapy, Pain and Exercise Research Luebeck (P.E.R.L.), Universität zu Lübeck, 23562 Lübeck, Germany; l.luebke@uni-luebeck.de (L.L.); tibor.szikszay@uni-luebeck.de (T.M.S.); kerstin.luedtke@uni-luebeck.de (K.L.); 2Center of Brain, Behavior and Metabolism (CBBM), University of Luebeck, 23562 Lübeck, Germany; 3Institute of Medical Informatics, University of Lübeck, 23562 Lübeck, Germany; marcin.grzegorzek@uni-luebeck.de; 4Laboratory of Pain Research, Institute of Physiotherapy and Health Sciences, The Jerzy Kukuczka Academy of Physical Education, 40-065 Katowice, Poland; w.adamczyk@awf.katowice.pl; 5Division of Behavioral Medicine and Clinical Psychology, Cincinnati Children’s Hospital Medical Center, Cincinnati, OH 45229-3026, USA; 6Department of Knowledge Engineering, University of Economics in Katowice, 40-287 Katowice, Poland

**Keywords:** chronic back pain, machine learning, autonomic nervous system, skin conductance, experimental heat stimulation

## Abstract

Multiple attempts to quantify pain objectively using single measures of physiological body responses have been performed in the past, but the variability across participants reduces the usefulness of such methods. Therefore, this study aims to evaluate whether combining multiple autonomic parameters is more appropriate to quantify the perceived pain intensity of healthy subjects (HSs) and chronic back pain patients (CBPPs) during experimental heat pain stimulation. HS and CBPP received different heat pain stimuli adjusted for individual pain tolerance via a CE-certified thermode. Different sensors measured physiological responses. Machine learning models were trained to evaluate performance in distinguishing pain levels and identify key sensors and features for the classification task. The results show that distinguishing between no and severe pain is significantly easier than discriminating lower pain levels. Electrodermal activity is the best marker for distinguishing between low and high pain levels. However, recursive feature elimination showed that an optimal subset of features for all modalities includes characteristics retrieved from several modalities. Moreover, the study’s findings indicate that differences in physiological responses to pain in HS and CBPP remain small.

## 1. Introduction

Pain as a complex phenomenon protects the body from actual or potential tissue damage [1]. The International Association for the Study of Pain (IASP) defines pain in this context as “an unpleasant sensory and emotional experience associated with, or resembling that associated with, actual or potential tissue damage” [2]. Pain and the Autonomic Nervous System (ANS) have a close anatomical and functional connection [3,4,5,6]. On the one hand, pain alerts us to damage to our body [7], as the unpleasant sensation inevitably attracts attention. So, we are urged to inhibit it or avoid related actions. On the other hand, pain triggers many unconscious changes and processes in the body. An essential aspect of this relationship is shaped by the influence of pain on the body, which causes various physiological responses in the ANS parameters [8]. Painful stimuli may evoke different physiological body responses such as changes in heart rate variability [9], respiratory rate [10] or skin conductance [11,12,13]. Vice versa, changes in these autonomic reactions can influence pain responses [14]. More specifically, the autonomic body responses are caused by noxious stimuli and not by the pain itself [15]. Pain in medical or clinical settings is usually assessed via subjective pain ratings, e.g., using a Visual Analogue Scale (VAS) [16]. These subjective pain ratings are challenging in treating people who cannot adequately express their pain, for example, people with neurological diseases, cognitive impairments, patients receiving intensive care or persons with changes in pain perception due to chronic pain. Chronic pain is characterized by central sensitization, which is associated with changes in pain behavior and pain perception [1,17,18,19]. A more objective pain assessment method would be helpful for the pain management of people with chronic pain or other mentioned impairments [15]. Previous research assumed that chronic pain is associated with autonomic dysfunction [20,21,22]. Additionally, most of these single measurements show alterations as a response to acute or chronic pain on a group level, but the variance in the data is always high. In contrast to clinical trials, experimental pain models might help understand the underlying aspects of pain disorders, such as chronic pain conditions. Therefore, this experimental study aims (1) to evaluate whether the combined data from various autonomic body responses of Healthy Subjects (HSs) and Chronic Back Pain Patients (CBPPs) can accurately predict the perceived intensity of experimental heat pain stimuli and (2) to exploratively compare the autonomic output of CBPPs with those of HSs.

## 2. Materials and Methods

### 2.1. Study Design and General Information

This experimental heat pain study was performed in a laboratory at the University of Lübeck. The study was conducted in accordance with the Declaration of Helsinki and agreement from the Ethics Committee of the University of Lübeck (reference number 19-262). The study was preregistered at the German Clinical Trials Register (DRKS00025295). Data are reported according to the *STROBE* statement [23]. To exploratively compare the autonomic output of CBPP during experimental heat pain stimulation with the responses of healthy participants, we used data from the PainMonit Database (PMDB) [24]. Participants were recruited by distributing recruitment flyers from June 2021 to May 2022.

### 2.2. Participants

HSs were eligible to participate if they stated they were healthy and pain-free on the day of examination. CBPPs were eligible to participate if they had chronic back pain lasting more than three months in the last two years. Participants had to be aged between 18 and 65 years. They were excluded if they had comorbidities affecting the nervous system, cardiovascular diseases or psychiatric illnesses requiring treatment or medication, dermal diseases at the non-dominant forearm, other diagnosed comorbidities requiring systematic drug consumption or if they were pregnant.

### 2.3. Outcome Measures

Previous research has evaluated various markers for their use in automated pain assessment [25], ranging from ANS markers such as Heart Rate Variability (HRV), Electrodermal Activity (EDA) or pupillary reflexes to biopotentials such as neuronal signaling, and neuroimaging such as monitoring brain cell activity using Position Emission Tomography (PET) scans. In particular, sensory systems to record responses from the ANS have been popular because these modalities are relatively easy to acquire (especially compared to neuroimaging or neural signalling methods). For example, the publicly available BioVid Heat Pain Database (BVDB) introduced by Walter et al. [26] in 2013 enabled researchers to investigate EDA, Electrocardiogram (ECG), and Electromyogram (EMG) changes in response to heat stimuli in healthy subjects. One of the findings, supported by multiple sources, is that skin conductance is sensitive to thermal stimuli and has been identified as the best single channel for automated pain detection [27,28,29,30], even to the point where fusing more modalities reduces classification rates [31]. This relationship between thermal stimuli and EDA has also been validated separately by different researchers. Another study [32,33] used thermal grills, which use interlaced tubes to evoke an illusion of pain by presenting cool and warm stimuli simultaneously. A high contribution of several properties of EDA to the performance of pain detection models and an increase in the EDA signal with higher stimulation were observed. Although significantly inferior to the EDA signal, ECG and EMG still contain valuable information for the pain recognition task and are expected to improve classification rates through dedicated fusion approaches [31,34]. In addition, the use of Blood Volume Pulse (BVP) was evaluated in separate studies to investigate its use in predicting pain in patients undergoing surgery [35]. An accuracy of 86.79% in a 2-class pain status classification revealed the relevant information of BVP for pain recognition. As acute pain increases the respiratory frequency, flow, and volume [10], it has been successfully incorporated into the fusion of physiological sensor modalities for pain detection before [36]. However, it is not as commonly used in the research community compared to the channels of the sensor mentioned above. Finally, Gouverneur et al. [37] showed that physiological time series data captured with wearable devices only may be sufficient for the detection of a level of pain as well.

Thus, indicators of autonomic responses to pain were measured via different wearable sensors (respiBAN Professional, Plux Wireless Biosignals S.A., Portugal; E4 Wristband, Empatica, United States) attached to the participant’s body. Both devices and the used adhesive electrodes are shown in Figure 1. The physiological responses were identified in dependence on the perceived pain intensity ratings of five different stimulus intensities measured via a Computerized Visual Analogue Scale (CoVAS) (Medoc, Ramat Yishai, Israel).

Skin Conductance (SC) was detected via self-adhesive electrodes attached to the medial phalanx of the index and middle finger of the non-dominant hand. The electrodes were connected to the respiBAN Professional (RB) device to record the EDA continuously during experimental heat pain stimulation. In addition, another EDA signal was measured by the Empatica E4 (E4) worn on the wrist of the non-dominant arm. For the measurement of HRV, three self-adhesive electrodes of the RB device were placed as an ECG attachment at the participant’s chest.

A second measure of cardiac activity was acquired by the E4 that records the BVP. The respiratory belt of the RB device was set at the participant’s chest to record respiration parameters during experimental heat pain stimulation. The piezoelectric sensor of the device measured the dilation of the belt during experimental heat pain stimulation. The signals were transferred to a computer and visualized as a respiratory wave. Self-adhesive electrodes were attached to the trapezius muscle (pars descendens) of the non-dominant body site to record neck muscle activity during experimental heat pain stimulation. The sensors of the RB device measured the incoming signals as an EMG.

In addition to the autonomic parameters, basic characteristics of the sample (gender, age, comorbidities) and data of the Pain Vigilance Awareness Questionnaire (PVAQ) German version, Pain Catastrophizing Scale (PCS) and Pain Health Questionnaire (PHQ-9) were collected. The PVAQ addresses the awareness, consciousness, vigilance and observation of pain [38]. The PCS contains questions about catastrophizing of pain [39], and the PHQ-9 addresses symptoms of depression [40].

### 2.4. Study Procedure

To guarantee a standardized study procedure and reduce bias, the study working group in Lübeck created a guideline for the data acquisition phase. Participants sat on a chair in a laboratory of the University of Lübeck and first had to fill out the digital questionnaires (PVAQ, PHQ-9, PCS) after they gave informed consent. The documents were pseudonymized and stored in a second room of the laboratory. Before starting the experimental heat pain stimulation, the devices and sensors were attached to the participant’s body.

To detect the individual Pain Tolerance (PT) and Pain Tolerance Threshold (PTT), a calibration phase was conducted before the main phase of the data acquisition. The CE-certified Contact Heat-Evoked Potential Stimulator (CHEPS; Medoc, Ramat Yishai, Israel) was attached to the central non-dominant forearm of the participant by a 20 mmHG inflated blood pressure cuff to standardize the contact between the skin and the thermode. Using the Medoc Software (version 6.4.0.22), the participants received manually applied heat pain stimuli of 10 s duration with increasing intensity, beginning at 40 °C and increased to a maximum of 49 °C in steps of 1 °C. The subjects were instructed to rate each stimulus on the CoVAS scale from “no pain” (=0/100) to “worst pain imaginable” (=100/100) as exactly as possible. PT and PTT were recorded on a separate Microsoft Office Excel list. After the calibration phase, there was a test run to check PT and PTT. For this, participants received one stimulus rated as not painful and one stimulus rated as most painful. If the test failed, the temperatures were adopted in the Microsoft Office Excel list, and the test run started again. This means that, e.g., if the first stimulus is rated painful on the CoVAS scale, although it has to be not painful, the temperature is reduced by the software. After identification of the individual PT and PTT, four painful stimuli temperatures were defined using the found thresholds and the following equation:(1)$$P_{i}=PT+\left(i\times R\right),$$
where i∈1,2,3,4 and R=(PTT−PT)/4. A non-painful stimulus was set below the painful threshold temperature given as NP=PT−R. The found values were adjusted in the main programme of the Medoc Software for each subject individually. To start the main experimental phase, the RB and E4 devices were connected to the working computer via Bluetooth. The participants were instructed to relax their non-dominant arms and fingers to avoid movement and, thus, artefacts in the sensor data. The probands subsequently received 40 heat pain stimuli of five different intensities in a randomized order. The stimuli lasted for 10 s with randomized breaks of 15–30 s between the stimuli. To avoid strong irritations of the skin, the thermode device was moved two centimeters distally or proximally after 20 stimuli. After 20 to 25 min, the experimental phase was completed, and all sensors were removed from the subject’s body.

### 2.5. Machine Learning Analyses

To evaluate the predictive power and to account for the differences between the two given datasets (PMDB dataset for HSs and *ChronPainMonit* dataset for CBPPs), the datasets were analysed using Machine Learning (ML) techniques. Individual steps in the pattern recognition chain were followed to create an automated pain classification system for each of the two datasets. First, the continuous time series data from the data acquisition were segmented. Specific characteristics, known as Hand-Crafted Features (HCF), were then extracted from the various sensor modalities. Finally, a Random Forest (RF) [41] model was trained and evaluated in a leave-one-subject-out (LOSO) cross-validation (CV). To estimate the impact of each feature, Recursive Feature Elimination (RFE) was applied to find the best feature set for the given pain classification task.

#### 2.5.1. Segmentation

ML systems rely on training datasets of data samples and associated labels. Following Gouverneur et al. [24], the continuous data streams of the data acquisition were segmented by cutting out areas of 10 s during the stimulus as painful windows. Moreover, the 10 s prior to each segment were used as non-painful baseline windows (B). NP and $P_{1}$–$P_{2}$ were applied as associated class labels during stimuli. In addition, CoVAS responses during the 2 s after the stimulus were also analyzed to capture late responses. Figure 2 illustrates the segmentation process, showing painful stimuli in red and non-painful baseline windows in green.

#### 2.5.2. Feature Extraction

According to [42], a variety of features, from simple statistical to sophisticated literature-based, were calculated separately for the E4 and RB EDA signals. For the ECG signal, the mean, standard deviation, and slope of the linear regression of RR intervals, Root Mean Square of the Successive Differences (RMSSD) and number of R peaks were retrieved. Inhalation and exhalation phases in the respiratory signal were computed using a trapezoidal detection [43], and characteristics, such as the number of phases or their mean amplitude, were extracted afterwards. For the EMG signal, an aggregation of the signal power spectrum was also performed. Finally, all available features for the corresponding sensor modality were extracted using *Neurokit 2* [44] (neurokit2==0.2.3) with *Python*. Finally, simple statistical features such as max, min, range, standard deviation, interquartile range, mean, local maxima and minima and the mean of the absolute value were calculated for all modalities. A total of 323 features were extracted for the various sensor modalities.

#### 2.5.3. Classification and Evaluation

For classification, RF models with 100 trees were trained on binary tasks such as B vs. $P_{4}$ for two main reasons. First, RFs have been shown to provide robust classification performance for a variety of ML tasks. Similarly, RFs perform significantly better in discriminating between no pain and high pain intensities in direct comparison to other traditional learning algorithms such as Support Vector Machines (SVMs) or even Neural Network (NNs) [24,34,45]. On the other hand, by their very nature, RFs allow a relatively straightforward interpretation of their decision process, especially in contrast to Deep Learning (DL), where networks often act as black boxes with only input and output being transparent. The tree-like structure of the forest’s underlying entities, individual Decision Trees (DTs), implements a hierarchical decision path [46]. At the top of the tree is a root, from which several internal nodes follow, applying a splitting function to the incoming data until a leaf of the model presents the predicted outcome. The overall complex prediction is thus broken down into minor problems based on individual features, which can be viewed and interpreted later to estimate a feature’s performance. Moreover, the models were evaluated using a LOSO CV, as it allows for fair evaluation of subject-specific data and is commonly used in the pain recognition field. In a LOSO evaluation, the performance is evaluated in several runs (folds). In each fold, one subject’s data are used as the test set, while the remaining data are used as the training set. This process is repeated so that each subject in the dataset has been part of the test set once. The overall performance is obtained by calculating the mean value of all folds’ performances. As data from the training set are never part of the test set, a realistic estimate of the classifier performance is estimated. The performance is given in terms of accuracy, as both evaluated datasets have balanced classes (equal number of samples per class), and it is the most commonly used metric in automated pain recognition systems.

To find an optimal set of features for HSs and CBPPs concerning the task of automated pain recognition, RFE was performed. It is an effective method for selecting predictive features [47] that has been applied to various ML tasks in the past. First, the classification performance is computed using all available features. By calculating feature importance from the impurity (sum of squared deviations around the node average) of the RF model [48], the least important feature of the set is identified and discarded. The classification system is then trained again on the newly found feature space. This step is repeated until only one feature remains. Usually, an optimal subset of features from the original dataset is found.

## 3. Results

### 3.1. Characteristics of Participants

Twenty-four CBPPs and 59 HSs were recruited from August 2020 to June 2022. After examination for eligibility, three CBPPs were excluded. From 59 HSs and 21 CBPPs, eight (seven HSs and one CBPP) were excluded due to technical problems during data acquisition. A single imputation was performed to deal with missing data in six subjects. Characteristics of participants of both groups are presented in Table 1.

### 3.2. Machine Learning

Table 2 summarizes the classification performance for HSs evaluated in a LOSO CV given in terms of accuracy. Each sensor modality alone and a combination of all are tested for each binary task, i.e., no pain vs. high pain (B vs. $P_{4}$).

Similar to previous publications [27,31,34,49], no pain vs. high pain remains the simplest task for the automated pain classification system yielding the best results. Other tasks, like no pain vs. medium pain or lower pain classes, remain challenging and gradually reduce the classification performance. For most sensor channels, only a performance around chance can be achieved for the task of no pain vs. low pain. In particular, the task of discriminating between no stimulus and a non-painful stimulus seems to pose problems for the learning algorithms. The results for this task remain around chance (50%), with the best results being achieved by EDA_E4 with an accuracy of 55.65%. In addition, the various sensor modalities differ greatly in performance results. The EDA signal (from the RB) yields the best classification performance scores with a maximum value of 91.70%, whereas the EDA derived from E4 gives slightly worse performance for all tasks (72.28% for B vs. $P_{4}$). Moreover, the ECG sensor performs relatively decent, giving a best performance of 61.63%. The classification systems based on other sensors (BVP, EMG and Respiration) seem to struggle and only make a random estimate. A combination of all sensors remains slightly worse than the EDA sensor (from the RB) alone with a best performance of 89.90%. In contrast, Table 3 shows the results obtained for the participants with chronic back pain. In general, the same paradigms can be found, and the outcomes do not differ greatly. However, there is a slight drop in performance for the best single modality (EDA_RB) to 89.58%. Moreover, the best performance achieved with the ECG signal dropped to 54.17%, while better results are achieved with the BVP signal, which now reaches up to 57.74% accuracy. There is also a slight increase in performance for the EDA signal derived from the E4, resulting in a performance of 74.40% for the B vs. $P_{4}$ task.

To further test whether the two cohorts differed in terms of physiological responses to pain, a small transfer study was also conducted. Table 4 summarizes the results for the RFs trained on the PMDB and evaluated on the CBPP data in a LOSO CV for the different tasks. Again, the results differ only marginally from those of the system trained on CBPP data alone (Table 3).

To find an optimal feature set for the task of automated pain classification, RFE was applied to the full feature space for the no pain vs. high pain task (as it is the most widely used one in the ML community). Figure 3 visualizes the results showing the classification performance in terms of accuracy for both datasets and the varying size of the feature space (x-axis). Each point of the graph represents a performance result given in accuracy for a LOSO CV. On the left side, the results of the original dataset can be found. Each subsequent point in the sequence resembles the performance result obtained by eliminating one feature from the existing feature space. Each test was performed five times to compensate for random variations in the performance of the RF models. Only the best result of each trial is presented here. The best performance of 93.62% with 15 features and 91.67% with 31 features could be achieved for the HS and CBPP datasets, respectively. In both optimal feature sets, most of the features are derived from the EDA signal collected at the finger site. While the model trained on healthy subjects focused on skin conductance characteristics derived from the RB, selecting only two respiration and 13 features from the EDA, the system selected a wider range of metrics from different sensors for the CBPPs. Here, three BVP, ten EDA (E4), 15 EDA (RB), one ECG and two EMG features were selected. The complete list of features is summarized in Table 5.

### 3.3. Autonomic Responses and Heat Pain Intensities

Moreover, box plots of the applied maximum thermode temperatures and maximum CoVAS ratings per dataset (HSs and CBPPs) and stimulus (NP to $P_{4}$) were generated to visually inspect variations in the calibration (and thus pain thresholds and pain tolerance thresholds) and the subjective pain ratings for the two groups. Figure 4 visualizes the applied thermode temperature for each individual stimulus separated by the two groups. Applied mean temperatures were slightly lower for NP and $P_{1}$ for CBPPs in comparison to the other group.

Furthermore, Figure 5 visualizes the maximum CoVAS rating during each individual stimulus separated by the two groups. The mean values of the CoVAS ratings were visibly slightly higher in CBPPs compared to the HSs dataset.

Similar to the analysis of individual CoVAS ratings and applied temperature stimulus values, several physiological responses to pain characteristics were also examined. The main characteristics of the pain level detection task for each modality (Table 5) were examined in more detail individually for the two groups. In other words, the last remaining feature for each sensor modality is shown for the HSs-based RFE procedure. Figure 6, Figure 7, Figure 8, Figure 9, Figure 10 and Figure 11 show boxplots for the heart rate derived from the BVP, the time when the maximum heart rate occurs computed by the ECG sensor, the range of the tonic component of the EDA derived from the E4, the time when the maximum value of the EDA measured by the RB is reached, the mean absolute values of the first differences of the standardized EMG signal, and the minimum values of the respiration signal, respectively. In comparison to Figure 4 and Figure 5, the physiological properties were also calculated and visualized for the segments of the baseline windows.

## 4. Discussion

This experimental study investigated the autonomic nervous system-associated output of healthy subjects and chronic back pain patients during experimental heat pain stimulation with individually calibrated stimulus intensities. ML techniques were used to evaluate the use of different sensor modalities for the task of automated pain detection in healthy subjects and patients with chronic back pain. The negligible differences in accuracy between the two groups (Table 2 and Table 3) suggest that there is little difference in the physiological response to pain between healthy subjects and patients with chronic back pain, at least from the perspective of the learning algorithms and the given detection task. This assumption is further supported by the transfer learning results (Table 4), which show that models trained on healthy subjects can still be used on patients with chronic back pain. The varying dataset sizes could also explain the minor differences in performed accuracy between the two datasets. In addition, the in-group results provide further insight into the field of automated pain detection. In general, the performance results around random guess for the B vs. NP task indicate that there is no difference in the physiological outcomes for a resting and a non-painful state, suggesting that the success in discriminating no pain from pain for the other tasks is related to the pain responses themselves rather than the detection of physiological responses to an applied thermal stimulus. As has already been shown, tasks with larger margins between pain intensities (B vs. $P_{4}$) are more manageable for the learning algorithms to cope with than smaller ones (B vs. $P_{1}$). Skin conductance remains one of the most promising and successful modalities that can be used. The performance of models trained only on the EDA (RB) signal outperforms others and even a combination of all (Table 2 and Table 3). Only in the case of using the RF trained on HSs for the CBPPs group (Table 4), a fusion of all channels improves the accuracy compared to relying only on the EDA signal. The study with two different sources for this modality shows that the electrodes used and their placement are highly important. The better performance of the standard electrode site at the fingers (RB) compared to the use of the wrist band (E4) shows that the choice of electrode site has a significant impact on automated pain detection tasks. The fusion of various sensor modalities only improved results for tasks like B vs. $P_{1}$ or, in the case of transfer learning, further highlighting the dominance for characteristics in the skin conductance for the task at hand but indicating that an aggregation of various sensor sources can lead to improved models.

Exploratory analysis of maximum CoVAS values per stimulus (Figure 5) and applied stimulus temperatures revealed that CBPPs had slightly higher subjective pain intensity ratings in all stimulus levels and slightly lower temperatures for NP and $P_{1}$ stimuli (Figure 4), suggesting a lower initial pain threshold. Exploratory analyses of the features of the physiological responses that contribute the most to the classification outcomes (Figure 6, Figure 7, Figure 8, Figure 9, Figure 10 and Figure 11) show minimal differences between the two groups. A reduction in activity in the EDA signal in CBPPs could be introduced by the increased age of the group itself [50]. This slight reduction in EDA could also justify the reduced performance of the automated pain classification system on CBPPs in comparison to HSs. Here, a reduced overall activation yields less information to evaluate the subject’s level of pain. As previously shown [42], the EDA sensor presents simplistic characteristics for the automated recognition of pain. Basically, a rise in EDA is associated with pain, whereas a reduction or monotonic levels of skin conducted are related to non-painful segments. An overall minimized activation in the EDA signal could thus constitute the slightly lower performance of the EDA (RB) signal in chronic back pain patients compared to healthy subjects. Moreover, both box plots of the given EDA features show visible differences in the distribution for B and $P_{4}$, showing their use for the classification task. Generally, EDA activation seems to be associated with painful segments, as an increase in tonic range is visible for the highest pain class (Figure 8). Moreover, maximum EDA peaks are found in higher pain classes (Figure 9), indicating that a rise of skin conductance with a late peak value is associated with pain, whereas a decreasing EDA signal (with a maximum value at the beginning of the time window) is associated with non-painful windows. Finally, differences in respiratory and EMG features between HSs and CBPPs show deeper respiration in CBPPs and more changes in the EMG signal compared to healthy probands. No exploratory differences were found between the groups for the BVP and ECG sensors.

The results of the exploratory analyses suggest that the autonomic output of chronic back pain patients during experimental heat pain stimulation may not differ from the output of healthy subjects, except for respiration. Interestingly, patients with chronic back pain showed higher values than healthy controls. This may indicate that chronic back pain patients may have an increased respiratory function with deeper flow compared to healthy controls during experimental heat pain stimulation. A systematic review by Jafari et al. [10] supports this result. They investigated the effect of experimental and clinical pain on respiratory outcomes and reported evidence for an increased inspiratory rate and flow as an autonomic response to pain. Furthermore, one of the included clinical studies observed increased ventilation (in minutes) in chronic pain patients [10]. The authors hypothesized that an increased respiratory response to pain reflects one component of the fight-or-flight autonomic reaction, preparing the organism for action [10]. Additionally, the authors investigated the effects of respiration on pain; some of the included experimental studies found evidence that slow breathing, breath holding, or relaxation instructions reduced pain [10]. Further research on breathing exercises in treating chronic back pain supports this result [51]. Thus, the finding that chronic back pain patients showed increased respiratory function compared to healthy controls may indicate a higher awareness of this hypoalgesic effect in patients with chronic pain. The lack of further research about the underlying neurophysiological mechanisms for respiratory responses to acute or chronic pain indicates a great demand for future studies to explore this subject.

Furthermore, the slight differences either in CoVAS ratings or in stimulus intensities indicate that chronic back pain patients may not be more sensitive to experimentally induced heat pain stimulation than healthy controls. This is inconsistent with previous investigations: the study by Carriere et al. [52] reported an increased sensitivity of chronic low back pain patients to mechanical pain procedures. The cross-sectional study further suggested that pain catastrophizing and higher pain expectancies might lead to an altered pain sensitivity [52]. These suggestions are consistent with the findings of Meints et al. [53]), who identified an association between pain catastrophizing and pain sensitization. This might explain the lack of a difference between CoVAS ratings because the included chronic back pain patients had a consistently low PCS score. However, PCS scores significantly differed from those of healthy subjects. Results might be different in a sample showing higher PCS scores.

### Limitations

The generalizability of the current data is limited to healthy subjects and chronic low back pain populations. The results of this study provide a first impression of chronic back pain patients’ autonomic reactions compared to healthy controls. Since pain is still a subjective experience modulated by different internal and external factors [54,55,56] and the fact that chronic back pain is a heterogeneous pathology [17], the results of this experimental study might have been influenced by other pain-modulating factors. Additionally, due to exploratory analyses, it was not possible to control our calculations for covariates, as there were differences in baseline characteristics (e.g., age, PVAQ, PHQ-9) between the groups. Furthermore, slightly more female than male subjects participated in both groups. However, previous studies assume that a sex-specific influence on autonomic parameters is rather unlikely [57,58,59]. Due to the small sample size of CBPPs, subgroup analyses could not be performed. However, there was no statistically significant difference in the proportion of female participants between the healthy and chronic back pain groups. Therefore, no influence on the explorative between-group results is expected. A further limitation is the constraint of experimentally induced heat pain. There is a demand for future studies with greater sample size and other pain induction methods or clinical pain studies to obtain a clearer understanding of the complex underlying mechanisms and relationships of nociception, autonomic reactions, and chronic pain [17,60]. In addition, there is a need for further investigations on pain assessment methods and the potential of wearable devices [61].

## 5. Conclusions

In summary, this study compared ML models trained on several autonomic parameters to predict the level of pain in HSs and CBPPs. The datasets used include experimental heat pain stimulation and several physiological responses recorded by two wearable devices. RFs were trained to evaluate their performance in discriminating different levels of pain and to identify critical sensors and features for the classification task using RFE. Similar to the results of previous publications, superior classification rates were achieved by distinguishing between no pain and severe pain as opposed to lower levels of pain. While EDA remained the best-performing modality, RFE applied to all modalities suggests an optimal subset of features, including those derived from multiple sensor channels. Thus, future work should focus on efficiently fusing different modalities to increase automated pain classification rates for individual subject cohorts. Finally, the learning algorithms indicate that there appears to be little difference between the physiological responses of HSs and CBPPs to pain, as classification performances are comparable for both groups, except for respiration parameters. The results provide a first impression of the potential for developing a pain-monitoring instrument based on the ANS. More findings within this topic can be relevant to enhance the quality of chronic pain care and to develop further therapeutic and pain assessment methods.

## Figures and Tables

**Figure 1 sensors-23-08231-f001:**
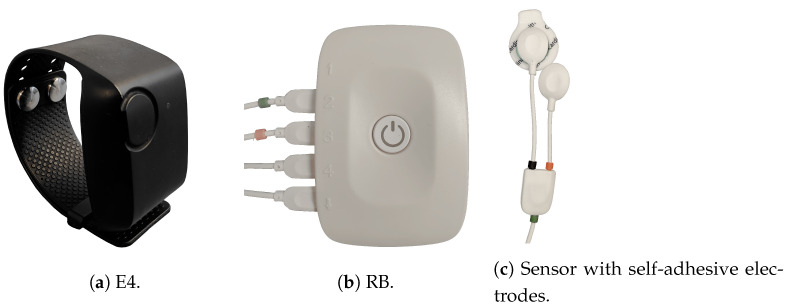
The wearable devices recording several physiological modalities during data acquisition of the PainMonit Database: (**a**) Empatica E4 (E4), (**b**) respiBAN Professional (RB), and (**c**) Electrodes used for the RB, i.e., Electromyogram.

**Figure 2 sensors-23-08231-f002:**
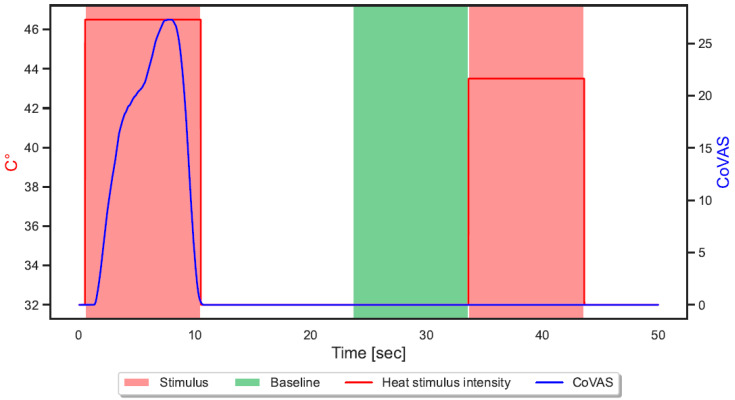
Visualization of the applied segmentation step on the retrieved data stream. The 10 s windows were extracted during the heat stimuli as painful windows (red) and before the stimulus as baseline windows (green).

**Figure 3 sensors-23-08231-f003:**
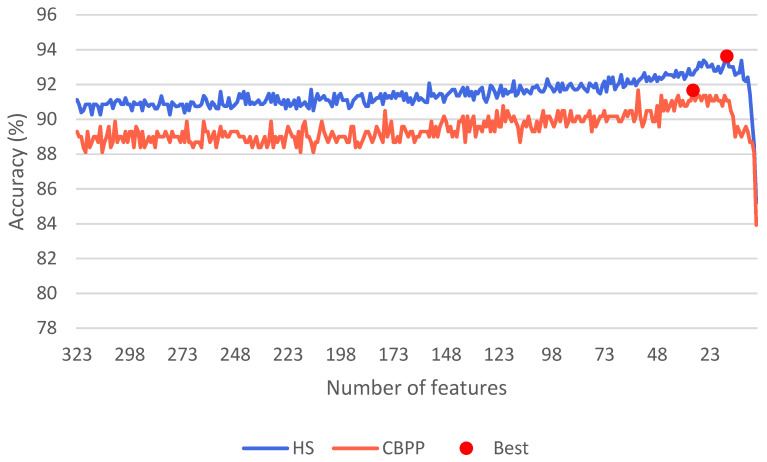
Comparison of the effect of different amounts of features (x-axis) on the resulting accuracy (y-axis) of a Random Forest evaluated on Healthy Subjects (HSs) and Chronic Back Pain Patients (CBPPs) in a leave-one-subject-out cross-validation for the no pain (B) vs. high pain ($P_{4}$) task.

**Figure 4 sensors-23-08231-f004:**
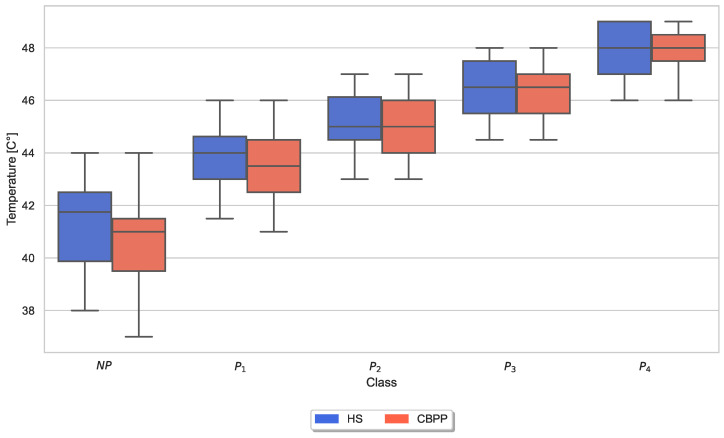
Box plots of applied thermode temperatures per dataset and stimulus. There are non-painful (NP) and four painful ($P_{1}$–$P_{4}$) stimulus temperatures. Values have been individually evaluated for Healthy Subjects (HSs) and Chronic Back Pain Patients (CBPPs).

**Figure 5 sensors-23-08231-f005:**
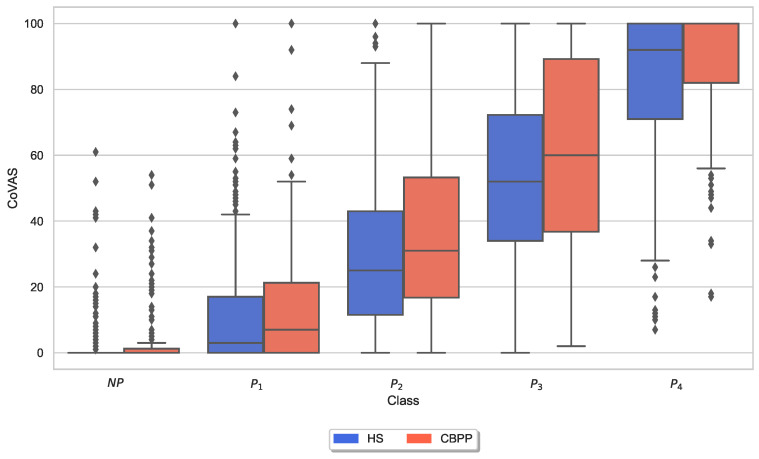
Box plots of the maximal Computerized Visual Analogue Scale (CoVAS) ratings per dataset and stimulus. There are non-painful (NP) and four painful ($P_{1}$–$P_{4}$) stimulus temperatures. Values have been individually evaluated for Healthy Subjects (HSs) and Chronic Back Pain Patients (CBPPs).

**Figure 6 sensors-23-08231-f006:**
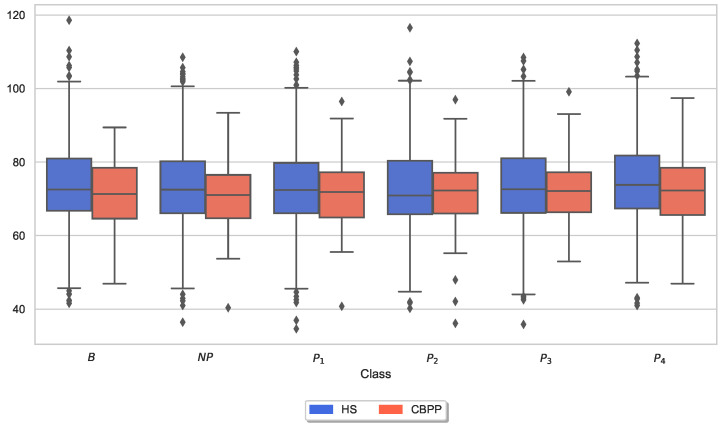
Box plots of the heart rate derived from the Blood Volume Pulse (BVP) per dataset and stimulus. There are non-painful (NP) and four painful ($P_{1}$–$P_{4}$) stimulus temperatures. Values have been individually evaluated for Healthy Subjects (HSs) and Chronic Back Pain Patients (CBPPs).

**Figure 7 sensors-23-08231-f007:**
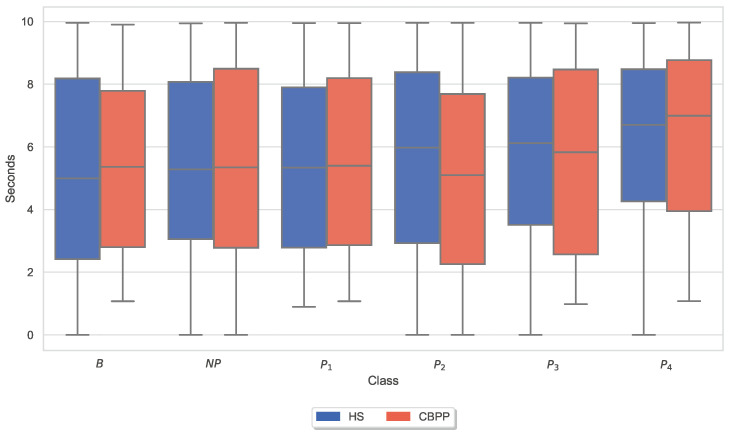
Box plots of the time when the maximum heart rate occurs computed by the Electrocardiogram (ECG) sensor per dataset and stimulus. There are non-painful (NP) and four painful ($P_{1}$–$P_{4}$) stimulus temperatures. Values have been individually evaluated for Healthy Subjects (HSs) and Chronic Back Pain Patients (CBPPs).

**Figure 8 sensors-23-08231-f008:**
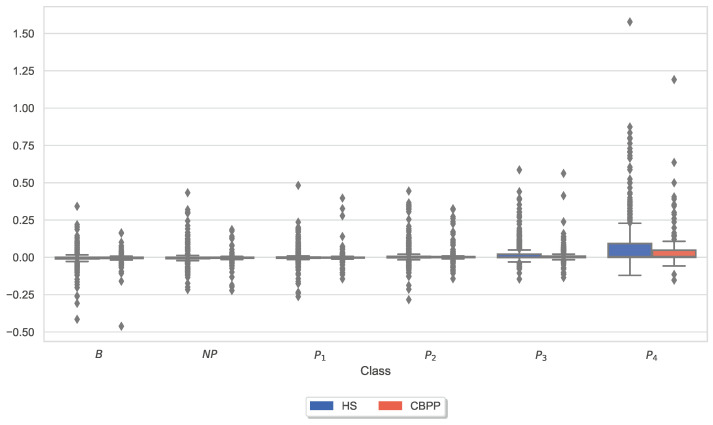
Box plots of the range of the tonic component of the Electrodermal Activity (EDA) derived from the Empatica E4 per dataset and stimulus. There are non-painful (NP) and four painful ($P_{1}$–$P_{4}$) stimulus temperatures. Values have been individually evaluated for Healthy Subjects (HSs) and Chronic Back Pain Patients (CBPPs).

**Figure 9 sensors-23-08231-f009:**
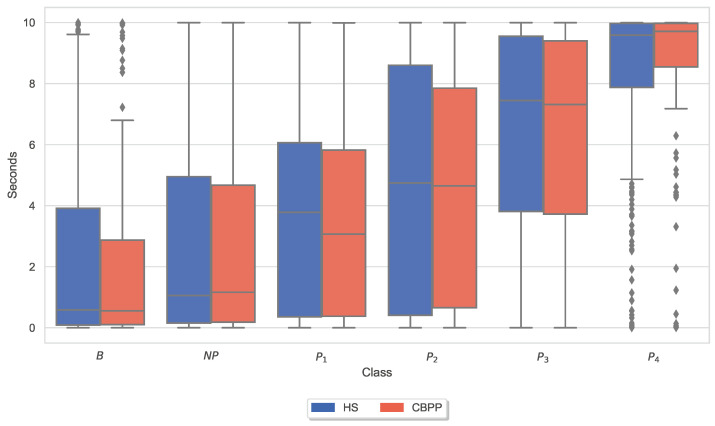
Box plots of the time when the maximum value of the Electrodermal Activity (EDA) measured by the RB is reached per dataset and stimulus. There are non-painful (NP) and four painful ($P_{1}$–$P_{4}$) stimulus temperatures. Values have been individually evaluated for Healthy Subjects (HSs) and Chronic Back Pain Patients (CBPPs).

**Figure 10 sensors-23-08231-f010:**
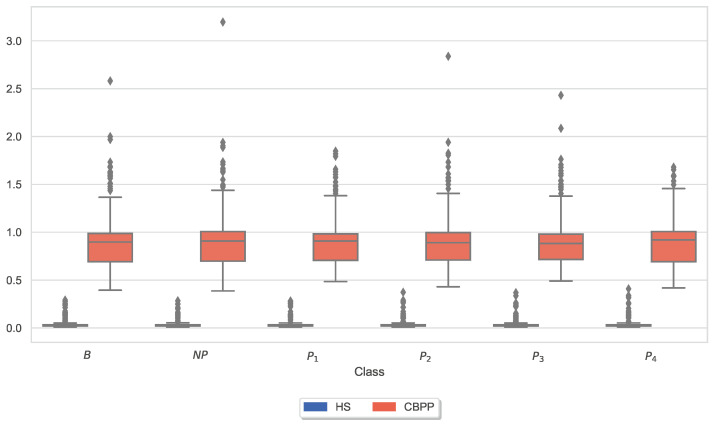
Box plots of the mean absolute values of the first differences of the standardized Electromyogram (EMG) signal per dataset and stimulus. There are non-painful (NP) and four painful ($P_{1}$–$P_{4}$) stimulus temperatures. Values have been individually evaluated for Healthy Subjects (HSs) and Chronic Back Pain Patients (CBPPs).

**Figure 11 sensors-23-08231-f011:**
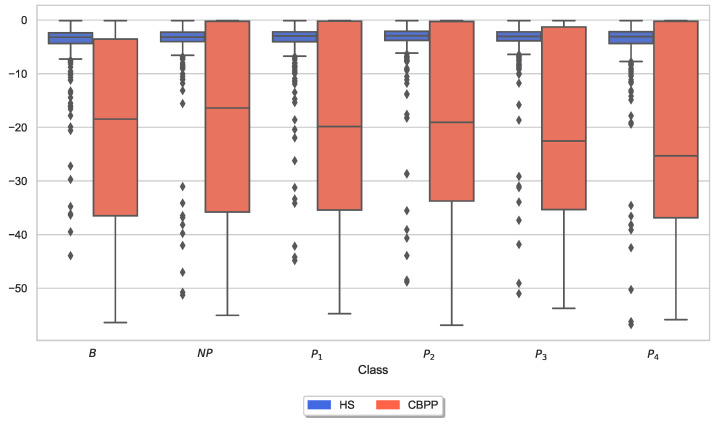
Box plots of the minimum values of the respiration signal per dataset and stimulus. There are non-painful (NP) and four painful ($P_{1}$–$P_{4}$) stimulus temperatures. Values have been individually evaluated for Healthy Subjects (HSs) and Chronic Back Pain Patients (CBPPs).

**Table 1 sensors-23-08231-t001:** Characteristics of study participants.

Characteristics	HSs (n = 52)	CBPPs (n = 20)	*p*
Age (Years), Mean (SD)	27.4 (6.6)	40.9 (14.4)	<0.001
Female, n (%)	35 (67.3)	15 (75)	0.534
BMI, Mean (SD)	23.4 (3.28)	24.7 (3.33)	0.106
PCS, Median (IQR)	14.0 (10.0)	12.0 (17.3)	0.015
PHQ-9, Median (IQR)	4.0 (2.3)	7.5 (7.0)	0.005
PVAQ, Median (IQR)	34.0 (10.0)	41.5 (18.8)	0.006

**Table 2 sensors-23-08231-t002:** Performance results (in %) of the Random Forest models trained on Healthy Subjects evaluated in a leave-one-subject-out cross-validation for several tasks given in accuracy.

Sensor	B vs. NP	B vs. $P_{1}$	B vs. $P_{2}$	B vs. $P_{3}$	B vs. $P_{4}$
Bvp	53.49	52.52	50.45	50.72	53.52
Ecg	52.16	45.79	51.63	54.33	61.63
Eda_E4	55.65	56.49	60.63	66.71	72.28
Eda_RB	50.84	61.78	68.11	78.12	91.70
Emg	50.36	52.28	49.21	48.20	52.02
Resp	51.08	48.32	52.80	53.12	54.82
All	52.64	62.26	67.24	77.76	89.90

**Table 3 sensors-23-08231-t003:** Performance results of the Random Forest models trained on Chronic Back Pain Patients evaluated in a leave-one-subject-out cross-validation for several tasks given in accuracy.

Sensor	B vs. NP	B vs. $P_{1}$	B vs. $P_{2}$	B vs. $P_{3}$	B vs. $P_{4}$
Bvp	47.32	49.11	55.95	50.89	57.74
Ecg	55.65	49.40	52.98	54.76	54.17
Eda_E4	49.70	50.60	56.25	58.93	74.40
Eda_RB	52.68	54.17	69.05	76.19	89.58
Emg	44.64	48.51	47.92	51.19	54.76
Resp	47.02	52.98	54.17	48.21	51.19
All	51.19	55.95	67.56	74.70	88.39

**Table 4 sensors-23-08231-t004:** Transfer performance results of the Random Forest models trained on the healthy subjects and evaluated in a leave-one-subject-out cross-validation on chronic back pain patients for several tasks given in accuracy.

Sensor	B vs. NP	B vs. $P_{1}$	B vs. $P_{2}$	B vs. $P_{3}$	B vs. $P_{4}$
Bvp	52.68	49.40	47.02	53.27	57.14
Ecg	50.30	53.27	48.81	43.75	53.27
Eda_E4	52.68	54.17	59.82	68.15	75.00
Eda_RB	53.87	56.25	66.96	77.98	87.80
Emg	51.49	51.19	47.62	48.81	46.73
Resp	49.11	53.57	55.65	48.21	56.25
All	51.79	58.33	65.48	76.19	88.10

**Table 5 sensors-23-08231-t005:** Optimal number of features found for the best accuracy of a Random Forest evaluated on Healthy Subjects and Chronic Back Pain Patients in a leave-one-subject-out cross-validation for the task no pain (B) vs. high pain ($P_{4}$) using Recursive Feature Elimination.

Dataset	# Features	Accuracy	Feature Set
CBPPs	31	91.67	’Bvp_Rate_Max_nk’, ’Bvp_Rate_Min_nk’, ’Bvp_Rate_SD_nk’, ’Eda_E4_diff_start_end’, ’Eda_E4_range_tonic’, ’Eda_E4_mean_rise_times’, ’Eda_E4_mean_offsets’, ’Eda_E4_norm_mean’, ’Eda_E4_dPhEDA_3’, ’Eda_E4_dPhEDA_10’, ’Eda_E4_TVSymp_6’, ’Eda_E4_TVSymp_7’, ’Eda_E4_SCR_RecoveryTime_nk’, ’Eda_RB_range’, ’Eda_RB_mean_abs_2_diff’, ’Eda_RB_argmax’, ’Eda_RB_argmin’, ’Eda_RB_diff_start_end’, ’Eda_RB_range_tonic’, ’Eda_RB_dPhEDA_3’, ’Eda_RB_dPhEDA_8’, ’Eda_RB_dPhEDA_9’, ’Eda_RB_dPhEDA_13’, ’Eda_RB_dPhEDA_14’, ’Eda_RB_dPhEDA_15’, ’Eda_RB_TVSymp_1’, ’Eda_RB_MTVSymp_1’, ’Eda_RB_SCR_RecoveryTime_nk’, ’Ecg_Rate_Baseline_nk’, ’Emg_mean_abs_1_diff’, ’Emg_SM3’
HSs	15	93.62	’Eda_E4_range_tonic’, ’Resp_min’, ’Resp_mean_in’, ’Eda_RB_max’, ’Eda_RB_min’, ’Eda_RB_iqr’, ’Eda_RB_argmax’, ’Eda_RB_argmin’, ’Eda_RB_diff_start_end’, ’Eda_RB_std_tonic’, ’Eda_RB_dPhEDA_3’, ’Eda_RB_dPhEDA_4’, ’Eda_RB_dPhEDA_6’, ’Eda_RB_dPhEDA_16’, ’Eda_RB_TVSymp_5’

## Data Availability

Data sharing does not apply to this article.

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
