# Peer review of "Objective Measurement of Subjective Pain Perception with Autonomic Body Reactions in Healthy Subjects and Chronic Back Pain Patients: An Experimental Heat Pain Study"

_sensors, 2023, doi:10.3390/s23198231_

Round 1

Reviewer 1 Report

The aim of this study is to evaluate whether the combination of 3 multiple autonomic parameters is more appropriate to quantify the perceived pain intensity of healthy subjects (HS) and chronic back pain patients (CBPP) during experimental heat pain stimulation. 

That is to say, it is to evaluate the selected autonomic parameters to see if they can quantify the perceived pain intensity. Therefore, there are two important aspects need to be addressed:

1. Selection of the autonomic parameters. Here, the authors did not present the design consideration of those "autonomic parameters", just simply adopting statistics.

2. Evaluation of those parameters as objective quantitative measures of perceived pain which is currently measured by subjective scales. That is, the evaluation is done by comparing "autonomic parameters" with subjective scales. It is typical clinical trial problems, but the authors use Machine Learning Analyses, RF models with 100 trees, without giving explanations of the choice.

Reviewer 2 Report

It is suggested the title be changed to include what the paper is about ‘experimental heat pain study’.

It is suggested that the specific sensors used to measure the outcoming need to be supported by reference to previous research where they were validated for pain measurement (add a brief discussion and references).

Line 105: delete ‘operated’ and add ‘applied’.

It is suggested that a brief discussion on why the various MI tools such as Random Forest were chosen and the advantages these provide to this particular study.

There is a gender bias in the data. It is suggested that a brief discussion as to how (or not) this bias may have affected the results.

It is suggested that the conclusion be expanded.

It is suggest the final manuscript be proof read by a nature English speaker.

Round 2

Reviewer 1 Report

NIL
